# Risk Factors for Postoperative Loss of Correction in Thoracolumbar Injuries Caused by High-Energy Trauma Treated via Percutaneous Posterior Stabilization without Bone Fusion

**DOI:** 10.3390/medicina58050583

**Published:** 2022-04-24

**Authors:** Ryosuke Hirota, Atsushi Teramoto, Hideto Irifune, Mitsunori Yoshimoto, Nobuyuki Takahashi, Mitsumasa Chiba, Noriyuki Iesato, Kousuke Iba, Makoto Emori, Toshihiko Yamashita

**Affiliations:** 1Department of Orthopaedic Surgery, Sapporo Medical University School of Medicine, Sapporo 060-8543, Japan; teramoto.atsushi@gmail.com (A.T.); myoshimo@sapmed.ac.jp (M.Y.); nobuyuki.takahashi0202@gmail.com (N.T.); 3203cb@gmail.com (M.C.); n.iesato@gmail.com (N.I.); iba@sapmed.ac.jp (K.I.); emrmkt@yahoo.co.jp (M.E.); tyamasit@sapmed.ac.jp (T.Y.); 2Department of Orthopaedic Surgery, Teine Keijinkai Hospital, Sapporo 060-8543, Japan; irifuneh@me.com

**Keywords:** thoracolumbar injuries, high-energy trauma, percutaneous posterior stabilization

## Abstract

*Background and Objectives*: Percutaneous pedicle screws were first introduced in 2001, soon becoming the cornerstone of minimally invasive spinal stabilization. Use of the procedure allowed adequate reduction and stabilization of spinal injuries, even in severely injured patients. This decreased bleeding and shortened surgical time, thereby optimizing outcomes; however, postoperative correction loss and kyphosis still occurred in some cases. Thus, we investigated cases of percutaneous posterior fixation for thoracolumbar injury and examined the factors affecting the loss of correction. *Materials and Methods*: Sixty-seven patients who had undergone percutaneous posterior fixation for thoracolumbar injury (AO classifications A3, A4, B, and C) between 2009 and 2016 were included. Patients with a local kyphosis angle difference ≥10° on computed tomography at the postoperative follow-up (over 12 months after surgery) or those requiring additional surgery for interbody fusion were included in the correction loss group (*n* = 23); the no-loss group (*n* = 44) served as the control. The degree of injury (injury level, AO classification, load-sharing score, local kyphosis angle, cuneiform deformity angle, and cranial and caudal disc injury) and surgical content (number of fixed intervertebral vertebrae, type of screw used, presence/absence of screw insertion into the injured vertebrae, and presence/absence of vertebral formation) were evaluated as factors of correctional loss and compared between the two groups. *Results*: Comparison between each group revealed that differences in the wedge-shaped deformation angle, load-sharing score, degree of cranial disc damage, AO classification at the time of injury, and use of polyaxial screws were statistically significant. Logistic regression analysis showed that the differences in wedge-shaped deformation angle, AO classification, and cranial disc injury were statistically significant; no other factors with statistically significant differences were found. *Conclusion*: Correction loss was seen in cases with damage to the cranial intervertebral disc as well as the vertebral body.

## 1. Introduction

Thoracolumbar injuries are among the most frequently occurring spinal injuries in younger individuals, most of which result from high-energy trauma, such as falls and traffic accidents [1,2]. Surgical treatment has classically involved internal fusion using open approaches [3]; however, recently, early operation after injury has been recommended based on the concept of spine damage control following high-energy trauma. Stabilizing the spinal column at an early stage of injury promotes the stability of hemodynamics and improvement of respiratory insufficiency, and it is said to both improve general condition and prevent complications.

In 2004, Assaker first reported the management of vertebral fracture via percutaneous fixation [4]. The technique has since been widely used due to its minimally invasive nature, proving an indispensable method for minimally invasive spine stabilization (MISt) of spinal trauma [5]. Percutaneous fixation for thoracolumbar injury has been reported to have good results [6], while correction loss and kyphosis are known to occur postoperatively in some cases, and few reports have shown long-term postoperative outcomes. We, therefore, investigated cases that underwent percutaneous posterior fixation for thoracolumbar injuries and followed for >12 months after surgery, and we examined the factors affecting the loss of correction.

## 2. Methods

The study protocol was approved by Institutional Review Board of each participating hospital. The present study was conducted in compliance with the Declaration of Helsinki, and the patients provided informed consent before participation. We included patients who underwent thoracolumbar fixation without bony fusion for unstable thoracolumbar injury. The operations were performed by the same surgeon between July 2009 and September 2016 at a single center. We performed thoracolumbar fixation using a percutaneous pedicle screw without bone grafting.

### 2.1. Patients’ Characteristics

Table 1 summarizes the general characteristics of the patient group. A total of 67 patients (average age, 49.4 years; male, *n* = 44; female, *n* = 23) who sustained injuries from high-energy trauma (traffic accidents and high falls), were transported to our advanced critical care center, and underwent surgery for unstable thoracolumbar injury (AO Spine Thoracolumbar Classification System: A3, *n* = 29; A4, *n* = 11; B1, *n* = 4; B2, *n* = 6; B3, *n* = 10, C, *n* = 7), and with a follow-up (FU) period of at least 1 year were eligible for this study. All patients had no neurological findings, such as muscle weakness or sensory disturbance, in the extremities. The exclusion criteria were concomitant malignancies with spinal metastases or any metabolic–endocrine comorbidities, pathological fractures, or osteoporotic vertebral fractures due to low-energy trauma in the elderly (>70 years). Cases with severe vertebral body destruction that required anterior column reconstruction at the time of admission were also excluded. After admission, patients underwent surgery as soon as possible, taking into account their general condition.

### 2.2. Surgical Procedure

Patients underwent short segment posterior fixation (2–6 levels; the extent of fixation was determined according to the degree of damage of the injured vertebra). The decision to perform vertebroplasty and intermediate screw insertion was made by the surgeon on a case-specific basis, and there were no strict criteria. Bone grafting was not performed.

### 2.3. Postoperative Procedure

In each case, patients were immediately allowed to move with a wheelchair. They were also allowed to ambulate depending on their pain and general condition after surgery, according to the pain experienced while wearing a thoracolumbosacral orthosis; the orthosis was worn for at least 3 months. At the follow-up visits at least 12 months after surgery, patients were radiographically evaluated. Patients with a local kyphotic angle difference ≥10° on computed tomography during postoperative follow-up, or those requiring additional surgery for interbody fusion, were referred to as the correction loss group.

### 2.4. Evaluation Items

Patients’ characteristics, sex, and age were investigated to determine the endogenous factors affecting outcomes. Radiologic characteristics including the level of vertebral injury, load-sharing score (LSS), AO-type classification [7], wedge-shaped deformation angle, local kyphotic angle (Figure 1), and degree of cranial and caudal disc injury at the time of injury were investigated to determine the endogenous factors affecting outcomes [8]. Operative characteristics including the number of fixations, type of pedicle screw (poly- or monoaxial), combination of vertebroplasty for injured vertebra body and intermediate screw insertion were assessed in the same manner.

### 2.5. Statistical Analysis

Univariate analysis was performed; for qualitative variables, Fisher’s exact probability test was used, while for quantitative variables, the Student’s *t*-test and Mann–Whitney U test were applied. A *p*-value of <0.05 was deemed statistically significant. Multivariate logistic regression analysis was also performed.

## 3. Results

Sixty-seven patients, with a mean age of 49.4 (range, 23–65) years and average follow-up duration of 20.4 (range, 13–40) months, were eligible. The mean waiting time to surgery was 2.6 (range, 0–10) days; mean operative time was 95.3 (range, 50–190) minutes. 

### 3.1. Surgical Content

The average fixed level was 2.7 (range, 2–6), while the average amount of blood loss was 124.7 (range, 20–300) mL. Polyaxial screws were used in 33 cases (49.3%), and monoaxial screws were used in 34 (50.7%). Vertebroplasty was performed in 18 cases (26.9%), whereas adding a screw to the fractured vertebra was performed in 44 cases (65.7%) (Table 2). There was no significant correlation between the number of fixed intervertebral segments and the addition of vertebroplasty or intermediate screw insertion (Table 3).There were no cases of significant intraoperative complications.

### 3.2. Correction Loss Evaluation

The average local kyphotic angle was 16.8° at the time of injury, which improved to 3.1° immediately after surgery, and had deteriorated to 9.5° by the final evaluation. The average loss of correction was 6.4°. We divided the subjects into two groups based on loss of correction, which was evaluated over 12 months after surgery. Loss of correction was considered positive when reoperation or a loss of local kyphotic angle >10° occurred; correction loss was observed in 23 cases, while 44 did not experience this loss. In the correction loss group, the average local kyphotic angle was 18.2° at the time of injury. This improved to 3.9° immediately after surgery and deteriorated to 16.1° by the final evaluation; the average loss of correction was 12.2°. In the no-loss group, the average local kyphotic angle was 15.1° at the time of injury, which improved to 2.7° immediately after surgery and deteriorated to 7.8° by the final evaluation; the average loss of correction was 5.1° (Figure 2).

### 3.3. Comparison between the Correction Loss and No-Loss Groups

Results of the univariate analysis indicated that the wedge-shaped deformation angle (*p* = 0.011), damage to the cranial disc (*p* < 0.01), load-sharing score (*p* = 0.030), AO classification (*p* < 0.01), and use of polyaxial screws (*p* = 0.029) exhibited significant differences between the correction loss and no-loss groups. The other explanatory variables including age (*p* = 0.775), sex (*p* = 0.503), local kyphotic angle (*p* = 0.245), caudal disc damage (*p* = 0.374), number of intervertebral fixations (*p* = 0.156), vertebroplasty (*p* = 0.526), and intermediate screw insertion (*p* = 0.391) were not significantly different between the two groups (Table 4). Next, we performed a multivariate logistic regression analysis, which showed that cases with an LSS score >7, compared with <6, had an 8.22-fold higher correction loss rate. When comparing AO classification cases, types A3, B2, or C exhibited a 3.88-fold increase in grade 3 damage to the upper disc, compared with grades 0–2; they also exhibited a 7.71-fold higher correction loss rate. These differences were statistically significant. No other factors were statistically significant (Table 5).

## 4. Discussion

In this study, we investigated the extent of postoperative correction loss and the risk for thoracolumbar injury treated only via percutaneous posterior fixation without bone grafting. We performed the surgery an average of 2.6 days after the injury, and encountered four cases (6.0%) that required anterior support reconstruction at a later stage; 19 cases (28.4%) suffered a loss of correction at >1 year postoperatively. Additionally, the degree of vertebral destruction (≥7 in LSS and type B2 and C fractures in AO classification) and intervertebral disc injury (grade 3: infraction of the disk into the vertebral body, annular tears, or herniation into the endplate) were independent risk factors for postoperative loss of correction; the degree of vertebral fracture was associated with correction loss, which is in line with previous reports. Some authors encourage the use of anterior support when the LSS score is ≥7 [9]. In this study, an LSS score ≥7 was also an independent risk factor for correction loss. Some studies have reported that ligamentous healing was mechanically weak, thus increasing the risk of instability (e.g., AO classifications B2 and C) [10]. In this study, this surgical procedure was performed in cases where it was judged possible to maintain a good repositioning position even with an LSS score of 7 or 8 points, taking bone quality and other factors into consideration. In recent years, relatively good results have been reported using posterior fixation even in cases with severe vertebral fracture [11]. However, in our study, severe vertebral injury was an independent risk factor for loss of correction. In addition to the degree of vertebral fracture, we found disc injury to be an independent risk factor for correction loss.

While it has been reported that end-plate injury causes apoptosis of disc cells and degeneration of the intervertebral disc, resulting in correction loss [12], it was also reported that more deformation resulted from the disc cavity than from the vertebral body [13]. Based on our results, it may be necessary to evaluate the extent of damage to the discs as well as the vertebral body; if the damage is severe, consideration should be given to performing anterior column reconstruction with tight intervertebral fusion. Although use of a polyaxial screw was not an independent risk factor for loss of correction, a significantly greater proportion of correction loss cases used polyaxial screws compared with no correction loss cases. Xue et al. reported that satisfactory fracture reduction and correction of segmental kyphosis can be both achieved and maintained via monoaxial pedicle screw fixation, including the fractured vertebra [14].

Recently, the use of percutaneous monoaxial screws has increased in popularity. We also recommend using this in severe vertebral fracture cases to add strong distraction and compression forces for kyphosis formation. Conversely, vertebroplasty and intermediate screw insertion did not prove advantageous in either the correction loss or no-loss groups. Evaluating the effectiveness of vertebroplasty and intermediate screw insertion purely for the prevention of correction loss is difficult, as in many cases, either one or the other were performed. Both vertebroplasty and injured vertebral screw insertion have been shown to effectively correct kyphosis [15,16]; however, it was not clear from this study which was more effective. Surgical treatment for thoracolumbar spinal injury is performed with the purpose of decompressing the nerves, realigning, and fixating the spine; the surgical approach is selected based on clinical symptoms, age, and injury type [17,18,19]. The posterior method has the advantage of being easier to deploy and learn as well as being less invasive when compared to the anterior approach. However, the most important theoretical advantage of percutaneous fixation of vertebral fractures is decreased bleeding and operative time [20]; it is relatively easy to perform surgery, even if the patient’s general condition is poor during the early postinjury period. In the present study, surgery was performed at an average of 2.6 days after injury, which may have contributed to the improvement in general condition, as well as early exercise allowance and discharge from the hospital.

Many authors have reported less blood loss during percutaneous fixation when compared with standard open fixation in their series [21,22]. Recently, the concept of spine damage control has been reported in the field of spine trauma, which is often accompanied by complications caused by high-energy trauma and based on the concept of damage control orthopedics in the treatment of multiple fracture trauma, such as iliac and pelvic ring fractures [23,24]. Early stabilization of the spinal column is expected to promote hemodynamic stability and improvement of respiratory failure, facilitate systemic management, and prevent complications. In particular, fixation using percutaneous pedicle screws (PPSs) has been shown to be useful as an early intervention for patients with relatively mild vertebral fractures. This less traumatic approach to the spinal muscle should theoretically result in decreased postoperative pain. Jiang et al. found better function and lower pain scores in the percutaneous group than in the open cohort [25], which is similar to the results reported in prior studies [26]. This allows patients to ambulate earlier, thereby being less exposed to bed rest complications and ulcers [27,28]. This would result in a significant decrease in patients’ hospital stay [22]. However, PPS fixation is not without its drawbacks. One of the drawbacks of this technique is that interbody fusion with bone grafting cannot be performed using only one technique; nevertheless, some reports suggest that bone grafting is not necessary.

Lyu et al. compared open instrumentation and fusion with open fixation and percutaneous fixation [29]. They concluded that percutaneous fixation alone, i.e., without grafting, is sufficient for treating these fractures. However, many reports suggest that residual kyphotic deformities have a poor prognosis and therefore recommend anterior strong brace reconstruction in cases of severe vertebral fractures with LSC > 7 [30]. Based on the results of the present study, in patients with Type B2 or C fractures (ligamentous injuries), or severe disc injury, as ligamentous healing is mechanically weak, thereby increasing the risk of instability, it should be noted that even with posterior instrumentation, correction loss may still occur. If the deformity progresses, anterior column reconstruction with intervertebral fusion may be an option, taking into account the back pain.

It has been reported that patients without neurological symptoms have better outcomes with conservative treatment regardless of kyphotic deformity [31], as well as that residual kyphotic deformities have a poorer prognosis [32].

In this study, we could not establish criteria for cases that needed long fusion, vertebroplasty, or intermediate screw insertion. In contrast, many studies have reported that in young patients with thoracolumbar spine injuries, vertebroplasty and intermediate screw insertions can help maintain the repositioned position [33,34,35,36,37]. Therefore, we believe that these techniques should be added to the treatment of thoracolumbar spine injuries except in cases with extreme destruction of the pedicle in which screw insertion may be risky. In the future, we plan to investigate whether vertebroplasty or intermediate screw insertion is more effective in preventing correction loss.

This study has several other limitations, including the fact that clinical outcomes were not evaluated. In our case group, evaluations were performed without implant extraction. Since the aim of this surgery is not to achieve intervertebral fusion, there are cases in which removal of the implant after vertebral body fusion may preserve the mobile segment; therefore, the loss of correction after implant removal should be evaluated in the future. Conversely, this study is appropriate for confirming the short-term outcomes of high-energy trauma injuries, as it evaluates the images of patients who have undergone percutaneous PPS fixation without bone fusion at a single center by the same surgeon, more than 1 year after surgery.

## 5. Conclusions

We investigated cases of percutaneous posterior fixation for thoracolumbar injury and examined the factors affecting the loss of correction; 67 patients who had undergone surgery for unstable thoracolumbar injury caused by high-energy trauma without neurological symptoms were eligible for this study. Correction loss was seen in cases with damage to the cranial intervertebral disc as well as to the vertebral body. In addition to percutaneous posterior fixation, anterior column reconstruction such as intervertebral fusion is recommended in severe cases of vertebral fracture or disc injury

## Figures and Tables

**Figure 1 medicina-58-00583-f001:**
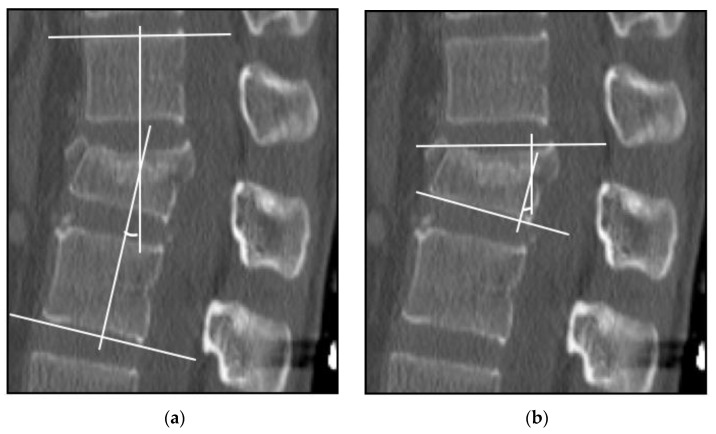
(**a**) Local kyphosis angle. (**b**) Wedge-shaped deformation angle.

**Figure 2 medicina-58-00583-f002:**
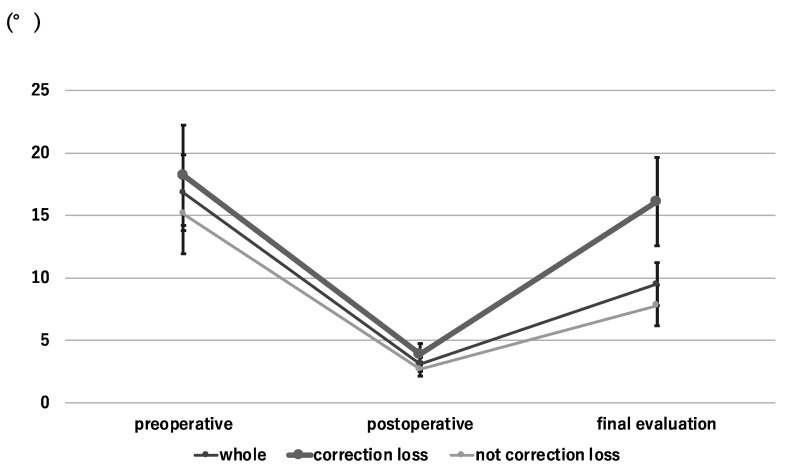
Change of the local kyphosis angle.

**Table 1 medicina-58-00583-t001:** Characteristics of the patients who underwent thoracolumbar fixation.

Number of Patients	67
Age (years)	49.4 ± 15.2
Male (%)	65.6
Follow-up period after fixation (months)	22.4 ± 9.7
Follow-up period after removal of implants (months)	8.5 ± 1.9
AO classification (%)	
A3	43.3
A4	16.4
B1	6.0
B2	9.0
B3	14.9
C	10.4
Level of injury (%)	
Thoracolumbar (Th9-L1)	71.6
Load sharing score	6.10 ± 1.85
Wedge-shaped deformation angle just after injury	15.0 ± 4.6°
Local kyphosis angle just after injury	16.8 ± 4.1°
Cranial disc injury just after injury (%)	
0–2	31.3
3	68.7
Caudal disc injury just after injury (%)	
0–2	83.7
3	16.3

**Table 2 medicina-58-00583-t002:** Surgical content.

Number of Fusions	3.4 (2–6)
Polyaxial screw/monoaxial screw (%)	49.3/50.7
Vertebroplasty (%)	26.9
Intermediate screw (%)	65.7

**Table 3 medicina-58-00583-t003:** Correlation between the number of fixed intervertebral segments and vertebroplasty or Intermediate screw insertion.

	Short (2–3) Segment Fusion Group *n* = 30	Long (4–6) Segment Fusion Group *n* = 37	*p*-Value
Vertebroplasty (%)	26.7	27.0	0.879
Intermediate screw (%)	70.0	62.2	0.623

**Table 4 medicina-58-00583-t004:** Comparison of characteristics of patients between the correction loss and no correction loss groups.

	Correction Loss (*n* = 23)	No Correction Loss (*n* = 44)	*p*-Value
Age	50.2 ± 16.6	48.5 ± 13.9	0.775
Sex (male %)	61.1	65.6	0.503
Level of vertebrae injury(thoracolumbar %)	68.9	72.7	0.462
Load sharing score	7.55 ± 2.11	5.20 ± 1.56	0.030
Wedge-shaped deformation angle	22.1 ± 6.2°	13.0 ± 4.6°	0.011
Local kyphosis angle	18.2 ± 3.9°	15.1 ± 3.1°	0.245
Cranial disc injury (%)			
0–2	43.4	4.5	<0.01
Caudal disc injury (%)			
0–2	82.6	86.4	0.374
AO classification (%)			<0.01
A3	26.1	52.3	
A4	13.0	18.2	
B1	8.7	4.5	
B2	17.4	4.5	
B3	13.0	15.9	
C	21.7	4.5	
Number of fixations	3.6 ± 1.5	3.3 ± 1.3	0.156
Pedicle screw (polyaxial pedicle screw %)	60.9	43.2	0.029
Vertebroplasty (%)	25.1	29.5	0.526
Intermediate screw(%)	69.3	63.6	0.391

**Table 5 medicina-58-00583-t005:** Multivariate analysis.

	Odds Ratio	95% CI	*p*-Value
Load sharing score			
4–6	1		
≥7	8.22	0.20–16.55	0.032
Wedge-shaped deformation angle	1.56		0.347
Cranial disc injury			
Grade 0–2	1		
Grade 3	7.71	0.25–11.41	0.018
AO classification			
A3	1		
B2 or C	3.88	0.65–5.15	0.034
Polyaxial pedicle screw	1.24	0.99–2.37	0.416

## Data Availability

The data presented in this study are available on request from the corresponding authors.

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
