# Peer review of "Risk Factors for Postoperative Loss of Correction in Thoracolumbar Injuries Caused by High-Energy Trauma Treated via Percutaneous Posterior Stabilization without Bone Fusion"

_medicina, 2022, doi:10.3390/medicina58050583_

Round 1

Reviewer 1 Report

Reviewer #1:

thank you for your submission. I have some questions for you.

RESPONSE: Thank you very much for your kind feedback.

Q1. Don't you think your cases were not severe for surgery? I think the preoperative wedge deformation angle(15‘) and local kyphosis angle(16.8‘) don't look severe.

A1.

Thank you very much for this very important point. As Reviewer #1 said, this study was conducted on cases in which the degree of vertebral destruction was relatively mild and considered addressable only by posterior stabilization. On the other hand, all patients were injured by high-energy trauma, and surgery was performed because it was considered to have advantages in terms of damage control and early rehabilitation.

Q2. In your univariate analysis, why do you think the number of fixations did not affect outcomes? I think the longer fixation is, the less loss of correction will be. 

A2.

Thank you very much for this point. The extent of fixation in surgery is determined by the surgeon, but it is natural for fixation to be longer in cases of severe vertebral destruction. Longer intervertebral fixation in cases with severe damage may counteract the preventive effect of corrective loss. In the future, we would like to investigate whether different fixation intervertebral spacing affects correction loss in cases with similar vertebral fractures.

Q3. In conclusion, you suggested anterior surgery in severe cases of vertebral fractures or disc injury even though cases in your study don't have neurologic problems. These days, even if there is a neurological deficit, there is a tendency not to perform anterior surgery well, but are you really recommending extensive anterior surgery to prevent radiological loss of correction? Is there any other way?

A3.

Thank you very much for this very important point. In recent years, surgical techniques and devices have evolved, and many cases can be handled only with posterior surgery, even if the damage is severe. In this study, we mentioned that there are many cases of correction loss in severe cases of vertebral fracture and disc injury, and we promote anterior column reconstruction as an option, but we think that surgery should be performed posteriorly first, and then anterior surgery should be considered when kyphosis deformity worsens .

In such cases, back pain at the level of injury may also be a criterion for deciding whether or not to perform anterior surgery. The relevant text in the Discussion has been modified accordingly (page 7, lines 258-264).

“Based on the results of the present study, in patients with Type B2 or C fractures (ligamentous injuries), or severe disc injury, as ligamentous healing is mechanically weak, thereby increasing the risk of instability, it should be noted that even with posterior instrumentation, correction loss may still occur. If the deformity progresses, anterior column reconstruction with intervertebral fusion may be an option, taking into account the back pain.”

Thank you. 

RESPONSE: Thank you once again very much for your kind feedback.

Reviewer #2:

Reviewer has gone through the article “Risk factors for postoperative loss of correction in thoracolumbar injuries caused by high-energy trauma treated via percutaneous posterior stabilization without bone fusion”. Authors may address the following queries:

Thank you very much for the feedback. We have responded to all your comments below.

GENERAL COMMENTS

Q1.Introduction may need rewriting as it gives a historic overview which may suit better in discussion

A1.

Thank you very much for this point. We have corrected the Introduction section accordingly(page 1-2, lines 36-57).

“Thoracolumbar injuries are among the most frequently occurring spinal injuries in younger individuals, most of which result from high-energy trauma, such as falls and traffic accidents [1, 2]. Surgical treatment has classically involved internal fusion using open approaches [3]; however, recently, early operation after injury has been recommended based on the concept of spine damage control following high-energy trauma. Stabilizing the spinal column at an early stage of injury promotes stability of hemodynamics and improvement of respiratory insufficiency, and is said to both improve general condition and prevent complications.

In 2004, Assaker first reported the management of vertebral fracture via percutaneous fixation [4]. The technique has since been widely used due to its minimally invasive nature, proving an indispensable method for minimally invasive spine stabilization (MISt) of spinal trauma [5]. Percutaneous fixation for thoracolumbar injury has been reported to have good results [6], while correction loss and kyphosis are known to occur postoperatively in some cases, and few reports have shown long-term postoperative outcomes. We, therefore, investigated cases that underwent percutaneous posterior fixation for thoracolumbar injuries and followed for >12 months after surgery, and we examined the factors affecting the loss of correction.”

REBUTTAL: Accepted

Q2.Reviewer feels the methodology may have limitations/shortcomings. There is no independent comparison of short segment vs long segment with and without intermediate screws.

A2.

Thank you very much for this very important point. We were evaluating the presence of intermediate screw insertion, but described screw insertion into damaged vertebrae. We have corrected the description to intermediate screw insertion. The presence or absence of an intermediate screw was not a significant orthodontic loss factor.

REBUTTAL: What reviewers suggested was that comparison should be made of use of intermediate screws in short vs long segment to understand whether any of these hold the advantage over the other .

Q3. Reviewer would want to point that the findings and conclusions, in regards to load sharing classification, disc injuries as the cause of failure is already established in literature in depth. There is nothing new in regards to the findings.

A3.

Thank you very much for this very important point. We fully agree with Reviewer #2. In this study, we were unable to identify any factors other than those that have been previously reported to cause new correction losses. On the other hand, there are few reports evaluating more than 60 cases of thoracolumbar spine injuries sustained from high-energy trauma by a single surgeon for more than one year, which we considered valuable.

REBUTTAL: Accepted and we agree that there is no new information from the study.

SPECIFIC COMMENTS

Q1. Authors may want to elaborate/reframe “without neurological status” in methodology. Does it mean that patients without neurological deficit were included? Or the neurological status is not known?

A1.

Thank you very much for this point. Our study included cases with no neurological deficit. We have modified the Materials & methods section accordingly to reflect this (page 2, lines 72-74).

“All patients had no neurological findings, such as muscle weakness or sensory disturbance, in the extremities”.

REBUTTAL: Accepted

Q2. Authors may want to define “high energy” as per the patients included?

A2.

Thank you very much for this point. In our study, we defined high-energy trauma cases as cases of injury from a traffic accident or high fall that were transported to our advanced critical care center. We have modified the Materials & methods section to reflect this (page 2, lines 68-70).

“who sustained injuries from high-energy trauma (traffic accidents and high falls), transported to our advanced critical care center,”.

REBUTTAL: Accepted

Q3. Authors may want to mention the indications of vertebroplasty in their cases as pathological fractures, malignancy, osteoporosis were excluded.

A3.

Thank you very much for this point. In this study, the decision to perform vertebroplasty and screw insertion into the damaged vertebrae was made by the surgeon on a case-specific basis and there were no strict criteria. We added a description regarding the decision of the surgical technique to the Materials & methods section (page 3, lines 91-93).

“The decision to perform vertebroplasty and intermediate screw insertion was made by the surgeon on a case-specific basis and there were no strict criteria.”

REBUTTAL: There are a lot of variables and there is no crisp comparison made among these 67 patients. Even if the decision to propose vertebroplasty and intermediate screw insertions are on a case specific basis in a scientific study there has to be some criteria to determine when it is to be used. Reviewer has reservations on the quality of the study.

Q4. Authors have included a category of AO B4 fractures. As per the reviewer, AO classifies type B fractures into three (B1, B2, B3) only. Authors may clarify.

A4.

Thank you very much for this point. We incorrectly listed C as B4 in Table 3. We have corrected the error accordingly.

REBUTTAL: Accepted

Q5. Authors may want to elaborate further on “In this study, an LSS score ≥7 was also an independent risk factor for correctional loss.” It is not mentioned in methodology whether in the primary surgery use of anterior column support was considered in such cases. Authors may want to justify with reasoning why it wasn’t considered?

A5.

Thank you very much for this very important point. In this study, this surgical procedure was performed in cases where it was judged possible to maintain a good repositioning position even with an LSS score of 7 or 8 points, taking bone quality and other factors into consideration. In recent years, relatively good results have been reported using the same technique even in cases with severe vertebral fracture [1]. We have added this content to the Discussion(page 6, lines 192-196).

“In this study, this surgical procedure was performed in cases where it was judged possible to maintain a good repositioning position even with an LSS score of 7 or 8 points, taking bone quality and other factors into consideration. In recent years, relatively good results have been reported using posterior fixation even in cases with severe vertebral fracture [11].”

REBUTTAL: Accepted

Q6. Authors may consider adding complications, if any, incurred in the surgical procedure.

A6.

Thank you very much for this point. There were no cases of significant intraoperative complications. We have added this information to the Results section (page 4, lines 130-131).

“The decision to perform vertebroplasty and intermediate screw insertion was made by the surgeon on a case-specific basis and there were no strict criteria.”

REBUTTAL: There are a lot of variables and there is no crisp comparison made among these 67 patients. Even if the decision to propose vertebroplasty and intermediate screw insertions are on a case specific basis in a scientific study there has to be some criteria to determine when it is to be used. Reviewer has reservations on the quality of the study.

Q7. In methodology authors have mentioned that these cases were managed by short segment fixation (2-6 levels). Authors may want to modify this as fixing 6 levels is not short segment fixation.

A7.

Thank you very much for this point. We have removed the phrase "short segment" in the surgical procedure under the Materials and methods section (page 3, line 90).

REBUTTAL: Accepted

Reviewer #3:

Basically, a solid workup of a manageable patient population is presented here.

RESPONSE:                                                              

Thank you very much for the feedback. We are very pleased to know that Reviewer #3 fully agree on the importance of our study.

The basic question is not new and no significant new findings are obtained. However, in my opinion, the presentation of a single surgeon collective is a good addition to the scientific literature 

RESPONSE:

Thank you very much for this very important point. We fully agree with Reviewer #3. In this study, we were unable to identify any factors other than those that have been previously reported to cause new correction losses. On the other hand, there are few reports evaluating more than 60 cases of thoracolumbar spine injuries sustained from high-energy trauma by a single surgeon for more than 1 year, which we consider valuable.

Reference

  1. Kanna, R.M.; Shetty, A.P.; Rajasekaran, S. Posterior fixation including the fractured vertebra for severe unstable thoracolumbar fractures. Spine J 2015, 15, 256-264.

Author Response

Thank you very much for this very important point. We were evaluating the presence of intermediate screw insertion, but described “screw insertion into damaged vertebrae”. We have corrected the description to “intermediate screw insertion”. The presence or absence of an intermediate screw was not a significant orthodontic loss factor.
REBUTTAL: What reviewers suggested was that comparison should be made of use of intermediate screws in short vs long segment to understand whether any of these hold the advantage over the other .
Response:
Thank you very much for this very important point. We examined whether there was a correlation between the number of fixed intervertebral segments and the addition of vertebroplasty or intermediate screw insertion and found no significant association. We have accordingly modified the results to reflect this change (page 4, lines 114–116) as follows and have also added Table 3.
“No significant correlation existed between the number of fixed intervertebral segments and the addition of vertebroplasty or intermediate screw insertion (Table 3).”

SPECIFIC COMMENTS 
 Q3. Authors may want to mention the indications of vertebroplasty in their cases as pathological fractures, malignancy, osteoporosis were excluded.
 A3. 
Thank you very much for this point. In this study, the decision to perform vertebroplasty and screw insertion into the damaged vertebrae was made by the surgeon on a case-specific basis and there were no strict criteria. We added a description regarding the decision of the surgical technique to the Materials & methods section (page 3, lines 79-81).
“The decision to perform vertebroplasty and intermediate screw insertion was made by the surgeon on a case-specific basis and there were no strict criteria.”
REBUTTAL: There are a lot of variables and there is no crisp comparison made among these 67 patients. Even if the decision to propose vertebroplasty and intermediate screw insertions are on a case specific basis in a scientific study there has to be some criteria to determine when it is to be used. Reviewer has reservations on the quality of the study.
Response:
Thank you for this comment. We completely agree with Reviewer #1. In this study, we could not identify cases for which vertebroplasty or intermediate screw insertions needed to be performed. This was because there were few cases for which none of these procedures were added. We have explained this in the Discussion(pages 7-8, lines 241–243) as follows:

“In addition, in this study, we could not establish criteria for cases that needed the addition of long fusion, vertebroplasty, or intermediate screw insertion, and this needs to be assessed in future studies.”

In contrast, it has already been reported in many studies that for thoracolumbar spine injuries in young patients, vertebroplasty and intermediate screw insertions can help maintain the repositioned position1-5). Therefore, we believe that these techniques should be added, except in cases of extreme destruction of the pedicle, which is risky for insertion. In the future, we plan to investigate whether vertebroplasty or intermediate screw insertion is more effective in preventing correction loss.

Q6. Authors may consider adding complications, if any, incurred in the surgical procedure.
 A6.
Thank you very much for this point. There were no cases of significant intraoperative complications. We have added this information to the Results section (page 4, lines 116-117).
“There were no cases of significant intraoperative complications.”
REBUTTAL: There are a lot of variables and there is no crisp comparison made among these 67 patients. Even if the decision to propose vertebroplasty and intermediate screw insertions are on a case specific basis in a scientific study there has to be some criteria to determine when it is to be used. Reviewer has reservations on the quality of the study.
Response:
Thank you for this important point. We completely agree with Reviewer #1. In this study, we could not identify cases for which vertebroplasty or intermediate screw insertions needed to be performed. This was because there were few cases for which none of these procedures were added. We have explained this in the Discussion(pages 7-8, lines 241–243) as follows:

“In addition, in this study, we could not establish criteria for cases that needed the addition of long fusion, vertebroplasty, or intermediate screw insertion, and this needs to be assessed in future studies.”

In contrast, it has already been reported in many studies that for thoracolumbar spine injuries in young patients, vertebroplasty and intermediate screw insertions can help maintain the repositioned position1-5). Therefore, we believe that these techniques should be added, except in cases of extreme destruction of the pedicle, which is risky for insertion. 
In the future, we plan to investigate whether vertebroplasty or intermediate screw insertion is more effective in preventing correction loss.

References
1) Toyone T, Tanaka T, Kato D, et al: The treatment of acute thoracolumbar burst fractures with transpedicular intra- corporeal hydroxyapatite grafting following indirect reduction and pedicle screw fixation: a prospective study. Spine. 2006; 31: E208-E214.
2) Toyone T, Ozawa T, Shirahata T, et al: Short-segment fixation without fusion for thoracolumbar burst fractures with neurological deficit can preserve thoracolumbar motion without resulting in post-traumatic disc de- generation. Spine. 2013; 38: 1482-1490.
3) Mahar A, Kim C, Wedemeyer M, Mitsunaga L, Odell T, Johnson B, et al. Short segment fixation of lumbar burst fractures using pedicle fixation at the level of the fracture. Spine. 2007; 32: 1503-1507.
4) Anekstein Y, Brosh T, Mirovsky Y. Intermediate screws in short segment pedicular fixation for thoracic and lumbar fractures: a biomechanical study. J Spinal Disord Tech. 2007; 20: 72-77.
5) Bolesta MJ, Caron T, Chinthakunta SR, Vazifeh PN, Khalil S. Pedicle screw instrumentation of thoracolumbar burst fractures: Biomechanical evaluation of screw configuration with pedicle screws at the level of the fracture. Int J Spine Surg. 2012; 6: 200-205.

Reviewer 2 Report

In my opinion, the quality of the paper has been further improved after the revisions have been made. In this form, I recommend publication without further changes. 

Author Response

Thank you very much for giving us the opportunity to strengthen our manuscript with your valuable comments and queries. We are thankful for the time and energy you expended. 

Round 2

Reviewer 1 Report

Thank you very much for this very important point. We were evaluating the presence of intermediate screw insertion, but described “screw insertion into damaged vertebrae”. We have corrected the description to “intermediate screw insertion”. The presence or absence of an intermediate screw was not a significant orthodontic loss factor.REBUTTAL: What reviewers suggested was that comparison should be made of use of intermediate screws in short vs long segment to understand whether any of these hold the advantage over the other .
Response: Thank you very much for this very important point. We examined whether there was a correlation between the number of fixed intervertebral segments and the addition of vertebroplasty or intermediate screw insertion and found no significant association. We have accordingly modified the results to reflect this change (page 4, lines 114–116) as follows and have also added Table 3.
“No significant correlation existed between the number of fixed intervertebral segments and the addition of vertebroplasty or intermediate screw insertion (Table 3).”

REBUTTAL: Accepted

SPECIFIC COMMENTS 
Q3. Authors may want to mention the indications of vertebroplasty in their cases as pathological fractures, malignancy, osteoporosis were excluded.
A3. Thank you very much for this point. In this study, the decision to perform vertebroplasty and screw insertion into the damaged vertebrae was made by the surgeon on a case-specific basis and there were no strict criteria. We added a description regarding the decision of the surgical technique to the Materials & methods section (page 3, lines 79-81).
“The decision to perform vertebroplasty and intermediate screw insertion was made by the surgeon on a case-specific basis and there were no strict criteria.”
REBUTTAL: There are a lot of variables and there is no crisp comparison made among these 67 patients. Even if the decision to propose vertebroplasty and intermediate screw insertions are on a case specific basis in a scientific study there has to be some criteria to determine when it is to be used. Reviewer has reservations on the quality of the study.
Response:
Thank you for this comment. We completely agree with Reviewer #1. In this study, we could not identify cases for which vertebroplasty or intermediate screw insertions needed to be performed. This was because there were few cases for which none of these procedures were added. We have explained this in the Discussion(pages 7-8, lines 241–243) as follows:

“In addition, in this study, we could not establish criteria for cases that needed the addition of long fusion, vertebroplasty, or intermediate screw insertion, and this needs to be assessed in future studies.”

In contrast, it has already been reported in many studies that for thoracolumbar spine injuries in young patients, vertebroplasty and intermediate screw insertions can help maintain the repositioned position1-5). Therefore, we believe that these techniques should be added, except in cases of extreme destruction of the pedicle, which is risky for insertion. In the future, we plan to investigate whether vertebroplasty or intermediate screw insertion is more effective in preventing correction loss.

REBUTTAL: Authors may consider mentioning in the limitations

Q6. Authors may consider adding complications, if any, incurred in the surgical procedure.
A6. Thank you very much for this point. There were no cases of significant intraoperative complications. We have added this information to the Results section (page 4, lines 116-117).
“There were no cases of significant intraoperative complications.”
REBUTTAL: There are a lot of variables and there is no crisp comparison made among these 67 patients. Even if the decision to propose vertebroplasty and intermediate screw insertions are on a case specific basis in a scientific study there has to be some criteria to determine when it is to be used. Reviewer has reservations on the quality of the study.
Response: Thank you for this important point. We completely agree with Reviewer #1. In this study, we could not identify cases for which vertebroplasty or intermediate screw insertions needed to be performed. This was because there were few cases for which none of these procedures were added. We have explained this in the Discussion (pages 7-8, lines 241–243) as follows:

“In addition, in this study, we could not establish criteria for cases that needed the addition of long fusion, vertebroplasty, or intermediate screw insertion, and this needs to be assessed in future studies.”

In contrast, it has already been reported in many studies that for thoracolumbar spine injuries in young patients, vertebroplasty and intermediate screw insertions can help maintain the repositioned position1-5). Therefore, we believe that these techniques should be added, except in cases of extreme destruction of the pedicle, which is risky for insertion. 
In the future, we plan to investigate whether vertebroplasty or intermediate screw insertion is more effective in preventing correction loss.

REBUTTAL: Authors may consider mentioning in the limitations

References
1) Toyone T, Tanaka T, Kato D, et al: The treatment of acute thoracolumbar burst fractures with transpedicular intra- corporeal hydroxyapatite grafting following indirect reduction and pedicle screw fixation: a prospective study. Spine. 2006; 31: E208-E214.
2) Toyone T, Ozawa T, Shirahata T, et al: Short-segment fixation without fusion for thoracolumbar burst fractures with neurological deficit can preserve thoracolumbar motion without resulting in post-traumatic disc de- generation. Spine. 2013; 38: 1482-1490.
3) Mahar A, Kim C, Wedemeyer M, Mitsunaga L, Odell T, Johnson B, et al. Short segment fixation of lumbar burst fractures using pedicle fixation at the level of the fracture. Spine. 2007; 32: 1503-1507.
4) Anekstein Y, Brosh T, Mirovsky Y. Intermediate screws in short segment pedicular fixation for thoracic and lumbar fractures: a biomechanical study. J Spinal Disord Tech. 2007; 20: 72-77.
5) Bolesta MJ, Caron T, Chinthakunta SR, Vazifeh PN, Khalil S. Pedicle screw instrumentation of thoracolumbar burst fractures: Biomechanical evaluation of screw configuration with pedicle screws at the level of the fracture. Int J Spine Surg. 2012; 6: 200-205.

Author Response

Reviewer #1

Thank you very much for the feedback again. We have responded to all your additional comments below.

SPECIFIC COMMENTS

Q3. Authors may want to mention the indications of vertebroplasty in their cases as pathological fractures, malignancy, osteoporosis were excluded.

A3. Thank you very much for this point. In this study, the decision to perform vertebroplasty and screw insertion for the damaged vertebrae was made by the surgeon on a case-specific basis, and there were no strict criteria. We have added a description regarding the decision of the surgical technique to the Materials & methods section (page 3, lines 79-81).

“The decision to perform vertebroplasty and intermediate screw insertion was made by the surgeon on a case-specific basis, and there were no strict criteria.”

REBUTTAL: There are a lot of variables and there is no crisp comparison made among these 67 patients. Even if the decision to propose vertebroplasty and intermediate screw insertions are on a case specific basis in a scientific study there has to be some criteria to determine when it is to be used. Reviewer has reservations on the quality of the study.

Response: Thank you for this comment. We completely agree with Reviewer #1. In this study, we could not identify cases for which vertebroplasty or intermediate screw insertions needed to be performed. This was because there were few cases for which none of these procedures were performed. We have explained this in the Discussion (pages 7–8, lines 236–243) as follows:

In this study, we could not establish criteria for cases that needed long fusion, vertebroplasty, or intermediate screw insertion….”

In contrast, many studies have reported that in young patients with thoracolumbar spine injuries, vertebroplasty and intermediate screw insertions can help maintain the repositioned position [33-37]. Therefore, we believe that these techniques should be added to the treatment of thoracolumbar spine injuries, except in cases with extreme destruction of the pedicle in which screw insertion may be risky. In the future, we plan to investigate whether vertebroplasty or intermediate screw insertion is more effective in preventing correction loss.

REBUTTAL: Authors may consider mentioning in the limitations

Response: Thank you very much for this very important point. As Reviewer #1 said, we have added a description of the surgical decision in the limitation (page 7, lines 236–244).

In this study, we could not able to establish criteria for cases that needed long fusion, vertebroplasty, or intermediate screw insertion. In contrast, many studies have reported that in young patients with thoracolumbar spine injuries, vertebroplasty and intermediate screw insertions can help maintain the repositioned position [33-37]. Therefore, we believe that these techniques should be added to the treatment of thoracolumbar spine injuries, except in cases of extreme destruction of the pedicle in which screw insertion may be risky. In the future, we plan to investigate whether vertebroplasty or intermediate screw insertion is more effective in preventing correction loss.

Q6. Authors may consider adding complications, if any, incurred in the surgical procedure.

A6. Thank you very much for this point. There were no cases with significant intraoperative complications. We have added this information to the Results section (page 4, lines 116–117).

“There were no cases with significant intraoperative complications.”

REBUTTAL: There are a lot of variables and there is no crisp comparison made among these 67 patients. Even if the decision to propose vertebroplasty and intermediate screw insertions are on a case specific basis in a scientific study there has to be some criteria to determine when it is to be used. Reviewer has reservations on the quality of the study.

Response: Thank you for this comment. We completely agree with Reviewer #1. In this study, we could not identify cases for which vertebroplasty or intermediate screw insertions needed to be performed. This was because there were few cases for which none of these procedures were performed. We have explained this in the Discussion (pages 7–8, lines 236–243) as follows:

In this study, we could not establish criteria for cases that needed long fusion, vertebroplasty, or intermediate screw insertion….”

In contrast, many studies have reported that in young patients with thoracolumbar spine injuries, vertebroplasty and intermediate screw insertions can help maintain the repositioned position [33-37]. Therefore, we believe that these techniques should be added to the treatment of thoracolumbar spine injuries, except in cases with extreme destruction of the pedicle in which screw insertion may be risky. In the future, we plan to investigate whether vertebroplasty or intermediate screw insertion is more effective in preventing correction loss.

REBUTTAL: Authors may consider mentioning in the limitations

Response: Thank you very much for this very important point. As Reviewer #1 said, we have added a description of the surgical decision in the limitation (page 7, lines 236-244).

In this study, we could not able to establish criteria for cases that needed long fusion, vertebroplasty, or intermediate screw insertion. In contrast, many studies have reported that in young patients with thoracolumbar spine injuries, vertebroplasty and intermediate screw insertions can help maintain the repositioned position [33-37]. Therefore, we believe that these techniques should be added to the treatment of thoracolumbar spine injuries, except in cases of extreme destruction of the pedicle in which screw insertion may be risky. In the future, we plan to investigate whether vertebroplasty or intermediate screw insertion is more effective in preventing correction loss.

This manuscript is a resubmission of an earlier submission. The following is a list of the peer review reports and author responses from that submission.

Round 1

Reviewer 1 Report

Dear Authors,

Thank you for your submission. I have some questions for you.

  1. Don't you think your cases were not severe for surgery? I think the preoperative wedge deformation angle(15‘) and local kyphosis angle(16.8‘) don't look severe.
  2. In your univariate analysis, why do you think the number of fixations did not affect outcomes? I think the longer fixation is, the less loss of correction will be. 
  3. In conclusion, you suggested anterior surgery in severe cases of vertebral fractures or disc injury even though cases in your study don't have neurologic problems. These days, even if there is a neurological deficit, there is a tendency not to perform anterior surgery well, but are you really recommending extensive anterior surgery to prevent radiological loss of correction? Is there any other way?

Thank you. 

Reviewer 2 Report

Reviewer has gone through the article “Risk factors for postoperative loss of correction in thoracolumbar injuries caused by high-energy trauma treated via percutaneous posterior stabilization without bone fusion”. Authors may address the following queries:

GENERAL COMMENTS

  1. Introduction may need rewriting as it gives a historic overview which may suit better in discussion
  2. Reviewer feels the methodology may have limitations/shortcomings. There is no independent comparison of short segment vs long segment with and without intermediate screws.
  3. Reviewer would want to point that the findings and conclusions, in regards to load sharing classification, disc injuries as the cause of failure is already established in literature in depth. There is nothing new in regards to the findings.

SPECIFIC COMMENTS

  1. Authors may want to elaborate/reframe “without neurological status” in methodology. Does it mean that patients without neurological deficit were included? Or the neurological status is not known?
  2. Authors may want to define “high energy” as per the patients included?
  3. Authors may want to mention the indications of vertebroplasty in their cases as pathological fractures, malignancy, osteoporosis were excluded.
  4. Authors have included a category of AO B4 fractures. As per the reviewer, AO classifies type B fractures into three (B1, B2, B3) only. Authors may clarify.
  5. Authors may want to elaborate further on “In this study, an LSS score ≥7 was also an independent risk factor for correctional loss.” It is not mentioned in methodology whether in the primary surgery use of anterior column support was considered in such cases. Authors may want to justify with reasoning why it wasn’t considered?
  6. Authors may consider adding complications, if any, incurred in the surgical procedure.
  7. In methodology authors have mentioned that these cases were managed by short segment fixation (2-6 levels). Authors may want to modify this as fixing 6 levels is not short segment fixation.

Reviewer 3 Report

Basically, a solid workup of a manageable patient population is presented here. The basic question is not new and no significant new findings are obtained. However, in my opinion, the presentation of a single surgeon collective is a good addition to the scientific literature